



# Monitoring planted forest expansion from 1990-2020 in China

Yuelong Xiao, Qunming Wang[*]

College of Surveying and Geo-Informatics, Tongji University, 1239 Siping Road, Shanghai 200092, China;

*Corresponding author*: Qunming Wang (wqm11111@126.com).

**Abstract**. China has undertaken extensive afforestation efforts in recent decades. However, the effectiveness of these plantings varies with different environmental conditions. Whether China's forest expansion is primarily due to intentional planting or natural reforestation remains uncertain. Thus, assessing the growth of planted forests (PF) is crucial for monitoring forest quality and supporting China's commitment to carbon neutrality. In this study, using 30 m Landsat time-series, we proposed a Continuous Change Detection and Classification (CCDC)-based PF expansion monitoring (C-PFM) method. Based on the C-PFM, 30 m annual maps for PF and natural forests (NF) across China from 1990 and 2020 were produced. The resulting PF map in 2020 achieved a F1-score of 79.2% for PF and an overall accuracy of 90.8% when validated against visually interpreted reference data. The PF maps for the years 1998, 2003, 2008, 2013, and 2018 were evaluated using data from the 5th, 6th, 7th, 8th, and 9th National Forest Inventory (NFI) data across 34 provinces and autonomous regions of China. The results demonstrated that all Pearson's product-moment correlations were larger than 0.86. According to the C-PFM results, we found 8.06 million ha (Mha) of net forest gains across China from 1990 to 2020, with 16.15 Mha net gains of PF and 8.09 Mha net loss of NF. In eight forestry ecological engineering areas, we observed that the upper and middle reaches of Yangtze river Shelterbelt Program and Pearl River Shelterbelt Program experienced the most significant PF expansion. The resulting dataset can serve as valuable scientific data for policymakers, researchers, and forest managers, guiding appropriate planting, environment enhancement, and carbon sequestration efforts. The produced 30 m annual maps for PF and NF in China are publicly available at https://doi.org/10.5281/zenodo.15559086 (Xiao, 2025).

## 1 Introduction

China has continued to expand planted forests (PF) to improve land cover, restore the environment, sequester carbon dioxide, and increase farmers' income (Yu et al., 2019; Dong et al., 2022; Lu et al., 2018). China is at the forefront of global greening efforts, contributing a 25% net increase in global vegetation despite covering only 6.6% of the world's vegetated area (Chen et al., 2019). Among the net





increase, 42% is attributed to the forest based on China's eight forestry programs, including the
Three-North Forest Shelterbelt Program, Taihang Mountain Greening Project, Liaohe River Shelterbelt
Program, Yellow River Shelterbelt Program, Huaihe River and Taihu lake Shelterbelt Program, upper
and middle reaches of Yangtze river Shelterbelt Program, Pearl River Shelterbelt Program, and Coast
Shelterbelt Program (Liu et al., 2023; Wang et al., 2007). Despite China's significant efforts and notable
achievements in increasing forest area, concerns are growing regarding the negative impact of PF
expansion.
The expansion of PF often occurs at the expense of replacing NF. However, previous studies have
found that PF cannot adequately replace NF due to their weaker resistance to disturbance, limited
ecosystem services, and lower biodiversity (Tang et al., 2007; Hua et al., 2022; Betts et al., 2022; Xu,
2011). In addition, PF in water-limited regions often exhibits lower survival rates and less effective
ecological and carbon sequestration outcomes compared to NF, largely due to singular planting patterns
and limited management practices (Cook-Patton et al., 2020). For example, plantation typically fails to
provide suitable habitat for other species and generates less organic matter in the soil, thus creating an
unsuitable living environment for soil organisms. Furthermore, the invasion of non-native tree species
disrupts the existing ecological balance, resulting in increased water consumption compared to native
species, which in turn leads to declining water levels and poses a threat to the survival of other
organisms (Xu, 2011). Planting suitable species at appropriate locations remains a challenge (Xu et al.,
2023), partially due to a lack of research on PF expansion monitoring.
Accurate mapping of PF and NF is critical for monitoring forest expansion, identifying the main
drivers of forest regrowth, and assessing carbon stock dynamics (Liao et al., 2024). It is also essential
for evaluating the ecological impacts of PF expansion on NF ecosystems (Fagan et al., 2022). However,
spatially explicit, long-term estimations of PF and NF expansion across China remain limited (Petersen
et al., 2016). Existing time-series products, such as those developed by Cheng et al. (2024), provide PF
and NF classification results at five-year intervals and a spatial resolution of 30. To generate time-series
training samples, Cheng et al. (2024) employed a monitoring approach that identified undisturbed PF
and NF pixels in 2020 and used them as training data for earlier years (i.e., 1990, 1995, 2000, …, and
2020). However, this method struggles in areas with frequent disturbances, as it lacks sufficient training
samples from disturbed regions. Some existing PF and NF maps focus on a single reference year and
emphasize data compilation. For example, Harris et al. (2019) developed the Spatial Database of
Planted Trees version 1 (SDPT_V1) by compiling country and region-level PF data. Bourgoin et al.
(2025) integrated several global datasets (such as SDPT, canopy height data from 2019, and primary




forest inventories) into a composite product (GFT2020) that classifies forests into three categories:
naturally regenerating forest, primary forest, and PF. However, such compilation-based datasets often
fail to capture the actual status of PF and NF in specific periods and typically suffer from variable
spatial resolution due to inconsistencies among data sources.
Thus, whether the growth of China's forested areas is driven by deliberate planting efforts or natural
reforestation remains unclear. Traditional methods of PF mapping primarily rely on manual delineation.
For example, Koskinen et al. (2019) delineated plantations in the southern highlands area of Tanzania.
However, manual delineation methods are time-consuming and labor-intensive, limiting their
implications at a large scale.
To date, utilizing machine learning to distinguish between PF and NF based on remote sensing
images become a more common choice. In general, machine learning methods are mainly performed
based on the differences in spectral, textural, and structural features between PF and NF. Recently,
various additional features have also been considered. For example, Cheng et al. (2023) fed the spectral,
temporal, structural, textual, and topographic features into a Random Forest (RF) classifier to map PF in
China. Fagan et al. (2018) utilized spectral, structural, and temporal features to train decision tree
models for mapping pine plantations in the southeastern U.S.. Koskinen et al. (2019) selected the
features extracted from optical and SAR images, and topographic data to train a classification and
regression tree (CART) classifier to map the plantation in the southern highlands area of Tanzania.
Additionally, the differences in phenology and management intensity between the PF and NF can also
be used as a criterion to identify these two kinds of forests. For example, Bey and Meyfroidt (2021)
utilized phenological and growth-based time-series attributes to distinguish plantations from natural or
semi-natural forests in North Mozambique. Deng et al. (2020) mapped the short-rotation eucalyptus
plantations of the Guangxi province of China based on the attribution that eucalyptus plantations should
be logged in a special period.
Although several studies have been developed to distinguish between PF and NF, methods
specifically designed for annual, time-series monitoring of PF and NF expansion remain scarce.
Previous studies mainly focus on differentiating two temporal PF maps to monitor PF expansion (Fagan
et al., 2022). This strategy can enlarge mapping errors and lead to cryptic forest loss (Puyravaud et al.,
2010). For example, errors in predicting the expansion of PF are magnified when using two PF maps
with misclassified areas due to errors in each classification. Consequently, it is difficult to assess
whether the forest regrowth is driven by PF or NF expansion, especially in China with rapid PF
regrowth. Furthermore, previous research cannot estimate the continuous PF expansion for a long





period instead of focusing on a single map in a certain year (Fagan et al., 2022).
To address the issues mentioned above, in this research, we proposed a Continuous Change Detection
and Classification (CCDC)-based PF expansion mapping (C-PFM) method to distinguish the PF and NF,
and more importantly, to monitor PF dynamic expansion annually. To achieve this goal, the C-PFM first
generates training sample points of PF and NF automatically. For each pixel, the corresponding
time-series is divided into multiple temporal segments by CCDC, with each segment sharing the same
fitted curve of the temporal profiles. Then, coefficients of the fitted curve from the last temporal
segment, covering 2020 at the training sample points, were selected as input features of the training data.
These samples are then fed into an RF classifier to classify other CCDC segments, which can
effectively filter out noise across different Landsat platforms. Finally, the classified segments are
transformed into annual PF and NF maps. The main contributions of this research are threefold: (1) The
C-PFM method was proposed to extract PF and identify its expansion at a national scale; (2) Annual 30
m resolution maps of PF and NF across China from 1990 to 2020 were generated; (3) An assessment of
the drivers of China's forest regrowth across eight forest ecological engineering areas and climatical
zones was reported. The results in this paper are crucial for advancing China's carbon neutrality goal
and enhancing ecosystem functions.
**2 Data**
**2.1 Remote sensing images**
All remote sensing images required for this study are available through the Google Earth Engine
(GEE) cloud platform (Tamiminia et al., 2020) at https://code.earthengine.google.com. We used Landsat
4-8 Collection 2 Tier 1 level 2 Surface Reflectance images from 1985 to 2020 across China for
time-series analysis to generate training samples. We enhanced image quality by removing shadows,
clouds, and snow pixels using the Quality Assessment band and performed atmospheric correction using
the LaSRC algorithm (Vermote et al., 2018). Any remaining gaps were filled using composite images
from adjacent years.
**2.2 The National Forest Inventory data**
In this paper, we selected 5th (1994-1998), 6th (1999-2003), 7th (2004-2008), 8th (2009-2013), and
9th (2014-2018) National Forest Inventory (NFI) (Zeng et al., 2023; State Forestry and Grassland
Administration of China, 2019) data to evaluate the accuracy of the produced PF and NF maps in five
years (i.e., 1998, 2003, 2008, 2013, and 2018). The areas of PF for 31 provinces (excluding Hong Kong,





Macao, and Taiwan) were reported by the NFI. As the Chongqing data are included in the Sichuan data
in the 5th NFI, only 30 provinces' records were used.

**2.3 Other existing PF and NF products**

Four existing PF and NF products were selected for comparison, as summarized in Table 1. Cheng et
al. (2024) provide a PF and NF classification dataset covering five periods (1990 to 2020, with a
five-year interval) at a spatial resolution of 30 m, which is publicly available for the entire China. Du's
planting year map was derived from the PF extent defined in SDPT_V1 (Harris et al., 2019; Du et al.,
2022), which was compiled based on PF data submitted by national agencies. SDPT_V2 is an updated
version of SDPT_V1, in which the PF data for China were replaced with a map at 1 km resolution
(Richter et al., 2024). SDPT_V2 represents the PF distribution for the year 2020. The GFC2020,
provided by the Joint Research Centre (JRC), is a global map of primary forests, naturally regenerating
forests, and PF for 2020 (Bourgoin et al., 2025). This product integrates multiple PF and NF datasets
through overlay analysis, incorporating disturbance information derived from the global forest cover
map with 30 m resolution. Additionally, the PFNF2021 (Xiao et al., 2024) and SBTN (Mazur et al.,
2025) datasets were employed to compare different training sample generation strategies. The
PFNF2021 dataset, developed by Xiao et al. (2024), provides a global map of PF and NF for the year
2021 at a spatial resolution of 30 m. The SBTN dataset serves as a 2020 baseline map of natural and
non-natural land covers, compiled from a synthesis of existing global and regional datasets (Mazur et al.,

143  2025).

Table 1. Summary of the existing PF and NF maps.

| ID | Name | Classes | Spatial resolution | Available years | References |
|----|------|---------|--------------------|-----------------|------------|
| 1 | Cheng's PF map | PF, NF | 30 m | 1990, 1995, 2000, 2005, 2010, 2015, 2020 | (Cheng et al., 2024) |
| 2 | GFC2020 | Primary forests, naturally regenerating forests, PF | / | 2020 | (Bourgoin et al., 2025) |
| 3 | Du's planting year map | PF | 1 km | 2020 | (Harris et al., 2019; Du et al., 2022) |
| 4 | SDPT_V2 | PF | 1 km (in China) | 2020 | (Richter et al., 2024) |
| 5 | PFNF2021 | PF, NF | 30 m | 2021 | (Xiao et al., 2024) |
| 6 | SBTN | NF | 30 m | 2020 | (Mazur et al., 2025) |



**2.4 Auxiliary data**

A forest structure database for plantation forests in China (CPSDv0) (Wu et al., 2023) was utilized as auxiliary data for visual interpretation to generate validation samples. The CPSDv0 includes information on tree species, mean stand age, mean tree height, stand density, and other attributes of plantations, derived from over 600 peer-reviewed articles.

When generating training samples and maps for PF and NF, we used the forest masks extracted from the WorldCover2020 land cover dataset (Zanaga et al., 2022) and the 30 m annual China Land Cover Dataset (CLCD) dataset (Yang and Huang, 2021). Produced by the European Space Agency, WorldCover2020 includes 11 classes: tree cover, shrubland, grassland, cropland, built-up areas, bare/sparse vegetation, permanent water bodies, herbaceous wetland, mangroves, and moss and lichen. We specifically extracted the tree cover class as the forest mask. In WorldCover2020, the tree cover class is defined as any geographic area dominated by trees with a canopy cover of $\geq$ 10%. This product has an overall accuracy of 74.4±0.1%, with the tree cover class having a user's accuracy of 80.8±0.1% and a producer's accuracy of 89.9±0.1%. At the continental level, WorldCover2020 has the highest overall accuracy in Asia at 80.7±0.1% (Tsendbazar et al., 2022). The forest mask derived from the WorldCover2020 dataset was utilized to identify training sample points, as its high spatial resolution provides more accurate information on forest locations. In addition, the CLCD dataset, currently one of the few sources providing annual land cover data for China from 1990 to 2020 at a 30 m resolution, was utilized to generate time-series forest masks. These masks were used for the post-classification refinement of the annual PF and NF extents identified in this study.

Additionally, we utilized shapefile data for eight forestry ecological engineering areas (Liu et al., 2023) and climate zones across China. The forestry ecological engineering shapefiles were obtained from the Geographic Data Sharing Infrastructure at the College of Urban and Environmental Science, Peking University (http://geodata.pku.edu.cn). The climate zone data was derived from the climate regionalization map of China (Zheng et al., 2010) and was digitized into a shapefile format.

**3 Methodology**

In this research, we proposed the C-PFM method to map annual PF maps. The workflow of the C-PFM approach is illustrated in Figure 1. In summary, the C-PFM method first employs the time-series analysis algorithm (i.e., CCDC) to identify training sample points for PF, NF, and non-forest. Next, features from the last CCDC segment, coupled with the labels, are selected and fed into the RF classifier.



The trained model is then used to predict all segments for all pixels across the study period from 1990 to
2020. Finally, the annual PF and NF maps are derived from all the classified CCDC segments. A
detailed description of the C-PFM method is provided below. Note that the entire process of mapping
annual PF forests was carried out on grids with 0.5 °×0.5 °(Xiao et al., 2023).

Figure 1. Flow chart of the C-PFM method for mapping annual PF and NF regrowth from 1990 to 2020.



### 3.1 Automatic training sample generation

We prepared training sample points with three labels: PF, NF, and non-forest. The training sample points were integrated from three sources. Specifically, NF sample points were identified by distinguishing differences in disturbance frequency between PF and NF. PF sample points were extracted from high-confidence areas, determined by integrating multiple PF maps in 2020. These maps included the four existing PF datasets listed in Table 1, as well as PF maps derived from a global NF and PF dataset produced by Xiao et al., (2023). To ensure the reliability of the PF samples, only areas where at least three PF products were consistent were used for extraction. Non-forest sample points were obtained from the non-forest mask of the WorldCover2020 dataset.

The NF training sample points were generated using the method developed in the study in Xiao et al., (2023). Specifically, PF exhibits higher disturbance frequencies than NF (Xu et al., 2024; Liao et al., 2012). The disturbance frequencies were calculated using the CCDC algorithm (Zhu and Woodcock, 2014; Xiao et al., 2023). Zhu and Woodcock (2014) proposed the CCDC algorithm for utilizing dense time-series of satellite imagery to detect changes and conduct land cover classification. In this research, a GEE-based CCDC (Arévalo et al., 2020) was applied to monitor the forest disturbance using all available Landsat images from 1985 to 2020. On the GEE platform, the key parameters of CCDC, *chiSquareProbability* and *minObservations*, which determine the sensitivity of disturbance detection, were set to 0.99 and 6, respectively (Xiao et al., 2023). Additionally, six spectral bands from the Landsat imagery—namely red, green, blue, near-infrared (NIR), shortwave infrared 1 (SWIR1), and shortwave infrared 2 (SWIR2)—were used in the CCDC for harmonic analysis. Additionally, we excluded sample points near open water to eliminate mislabeled samples influenced by frequent tides (Saintilan et al., 2022).

Based on the optimal training sample size for each tile (will be detailed in Section 4.3), a total of 1,343,709 training sample points were generated across China. These include 395,492 NF samples, 352,172 PF samples, and 596,045 non-forest training samples (Figure 2). To maintain class balance during model training, a fixed number of training samples per class was set for each 0.5 ° × 0.5 ° tile. When the available samples for a given class fell below this threshold, additional samples were sourced from adjacent tiles to ensure sufficient representation.



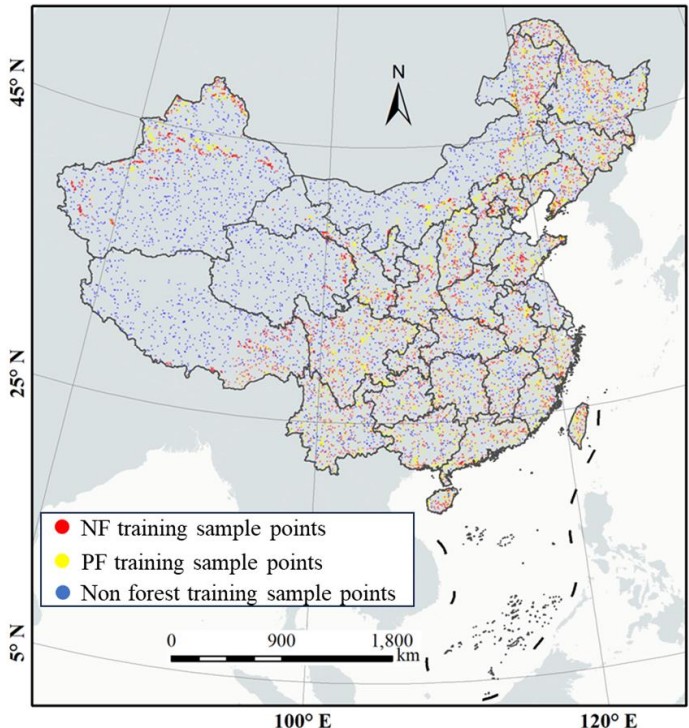


Figure 2. Spatial distribution of the 1,343,709 training samples across China.
**3.2 Classification of PF and NF based on CCDC segments**
**3.2.1 Selection of features**
For the training sample points, the features from the last CCDC segment (fitted curve) ending with
July 27, 2020, were selected as input features. To ensure the representativeness of features, segments
shorter than two years, spanning the period from 2019 to 2020, were excluded. The retained training
samples, combined with 261 features—including 6 spectral bands, 5 spectral indices (NDVI, NBR,
NDMI, EVI, and BSI), and 18 textural features multiplied by 9 coefficients (Figure 1) from the final
CCDC model—were used as inputs for the classifier. The feature importance score was evaluated using
the Gini coefficient calculated by the RF classifier (Cheng et al., 2023). To evaluate the influence of
dimensionality of the input, we tested four subsets of input features: the top 30 features (Features_30),
the top 60 features (Features_60), the top 90 features (Features_90, see Figure 3), and the full set of 261
features (Features_all).

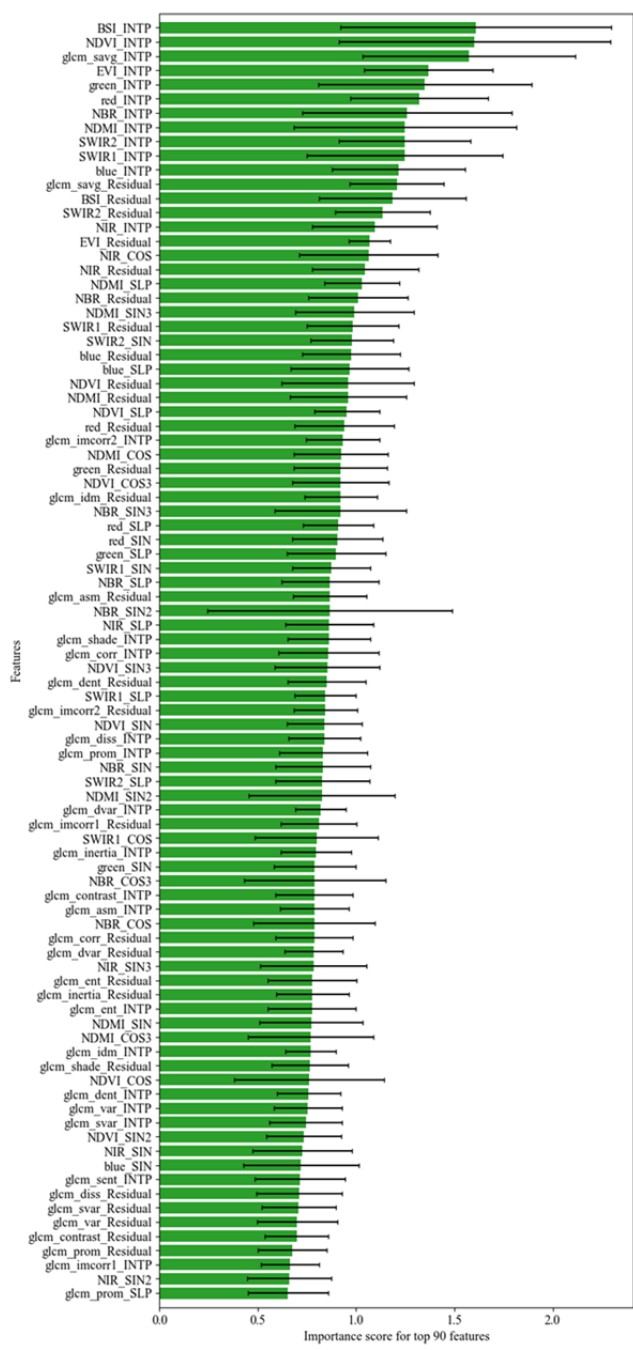


Figure 3. Importance scores of input features based on the Random Forest classifier. Only the top 90 features are

shown as examples. The feature names are consistent with 'band' names and coefficient names. For example,

'NBR_Residual' refers to the residual coefficient from the CCDC model for the NBR band. Features starting with

'glcm' represent CCDC coefficients for the 'bands' composed of textural features.



### 3.2.2 Classification based on the RF


Given China's vast latitudinal and topographic range, forest ecosystems span a wide array of climatic
zones—from the cold temperate zone in the north to subtropical and tropical climates in the south.
These climatic gradients substantially influence forest structure, phenology, and spectral characteristics,
thereby impacting classification accuracy. To address this issue, a localized RF model at a spatial extent
of $0.5\,° \times 0.5\,°$ was implemented. This approach helps mitigate the effects of inter-regional variability in
forest types and climatic conditions. RF classifiers are well-suited for classification tasks in land cover
mapping (Htitiou et al., 2021; Belgiu and Csillik, 2018). The trained classifier was applied to all CCDC
segments to estimate forest types for other periods. After identifying the forest types of all the segments,
we transformed them into annual maps of PF, NF, and non-forest. To ensure the quality of maps, the
pixels of maps in 2019 and 2020 were reclassified based on the method proposed in our previous study
(Xiao et al., 2024), as we excluded the training samples with shorter than two years in the last segments.
Additionally, a post-processing step was performed to reduce misclassification caused by sparse shrubs
and other vegetation that are easily confused with secondary forests. Specifically, forest masks extracted
from CLCD were applied to delineate the final forest extent for each corresponding period. Within this
extent, PF areas were first identified, and the remaining forest areas were classified as NF.

### 3.3 Accuracy assessment


### 3.3.1 Assessment with validation samples


To validate the accuracy of the produced map of PF in 2020, the validation sample collection was
designed by adopting the methodology recommended by Olofsson et al (2014). The AREA$^2$ (Area
Estimation & Accuracy Assessment) tool (available at https://area2.readthedocs.io/en/latest/), which
supports best practices for accuracy assessment and area estimation following Olofsson's framework,
was utilized. AREA$^2$ is implemented on the GEE platform (Bullock et al., 2020), which is compatible
with our mapping workflow that also runs on GEE. Within AREA$^2$, we applied stratified random
sampling. The parameters were set as follows: the target standard error for the PF class was set to 0.005,
the estimated user's accuracy was set to 0.95, and the anticipated proportion of PF within other classes
was set to 0.01. These settings were informed by recommendations in Adrah et al. (2025). Finally, the
total number of validation sample points was determined as 490. To ensure a minimum sample size for
each class, 50 samples were initially allocated to each. The remaining samples were then proportionally
distributed according to class area. After excluding several samples due to the unavailability of very
high-resolution Google Earth imagery, a total of 81, 67, and 319 samples were ultimately assigned to



NF, PF, and non-forest areas, respectively. We visually interpret the land cover labels based on the
locations of the generated validation sample points. In practice, non-forest labels are relatively
straightforward to interpret. To distinguish between PF and NF labels, several criteria were applied. For
PF interpretation, indicators such as regular planting patterns, evidence of management activities,
uniform canopy height, and consistent texture were considered. In contrast, NF is typically
characterized by greater diversity in canopy structure, color, and size, and lacks the regular features
associated with PF (Fagan et al., 2022). Figure 4 presents six representative examples of PF and NF
samples overlaid on high-resolution Google Earth images.

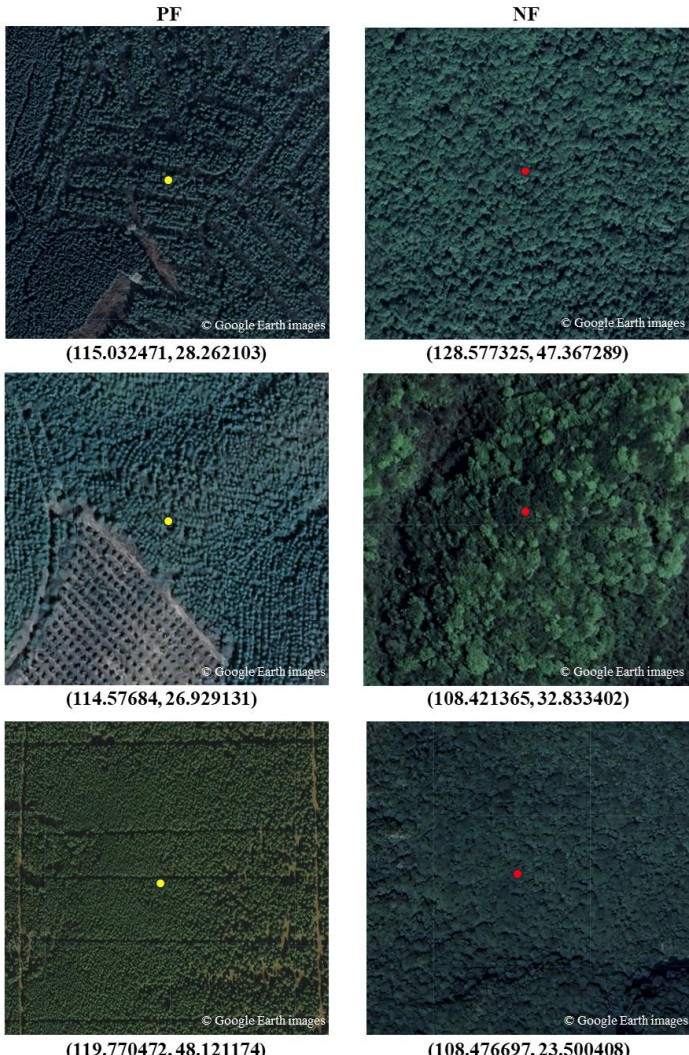


Figure 4. Six representative examples of PF and NF samples overlaid on high-resolution Google Earth images.
An independent set of time-series validation samples was incorporated to enable a more



comprehensive evaluation. Specifically, 490 validation sample points were stratified and randomly
generated across the testing areas (Figure 5). Each point was manually interpreted using historical
imagery available in Google Earth Pro, and reference labels were assigned based on observed land
cover changes. Due to the limited availability of high-quality historical imagery in earlier years, the
number of validation samples varied across periods. To ensure adequate representation, several early
years were aggregated into broader periods. Consequently, we obtained 89-337 validation samples for
the following time intervals: 2000–2006, 2007–2010, 2011–2012, 2013, 2014, 2015, 2016, 2017, 2018,
2019, and 2020.

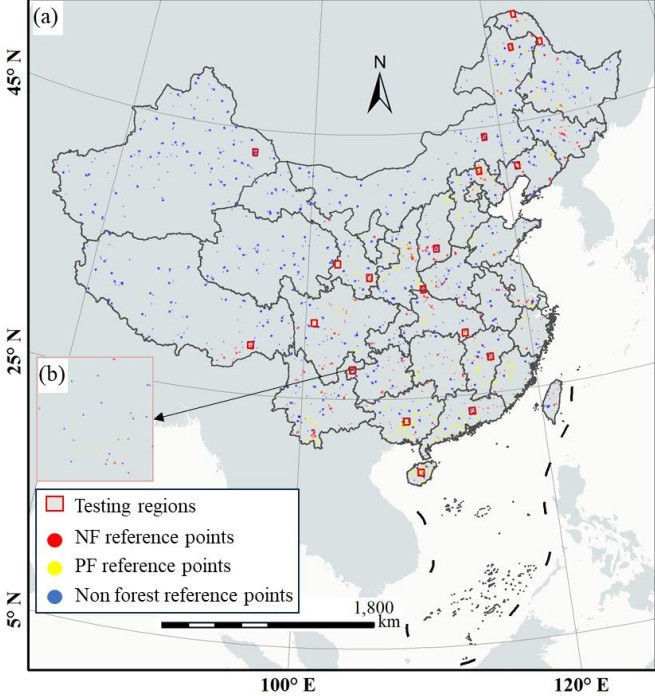


Figure 5. Spatial distribution of validation samples across China. (a) All validation samples; (b) Enlarged view of
validation samples in a testing region.
To better evaluate the performance across different regions of China, additional validation samples
are desirable. Thus, we adopted two sampling strategies: global sampling and map sheet-based sampling
(Tong et al., 2011; Chen et al., 2015). For the global sampling, the NF reference samples are randomly
distributed in China, and the PF reference samples were visually interpreted based on the locations of
PF sites in CPSDv0. Non-forest reference samples were visually interpreted according to randomly
generated points within the non-forest mask extracted from WorldCover2020. This strategy resulted in



2,116 global samples, comprising 485 NF samples, 1,035 PF samples and 596 non-forest samples. The
map sheet-based samples was generated in our previous study (Xiao et al., 2024). Specifically, China
was divided into 5 °×5 ° grids, with the centers of these grids serving as map sheets (0.5 °×0.5 °). We
excluded sheets with a low proportion of forest cover or lacking very high spatial resolution Google
Earth images. In all selected map sheets, there are 1,458 validation samples, including 250 NF samples,
PF samples and 864 non-forest samples. Eventually, we obtained a total of 3,574 additional
reference samples, including 735 samples for NF, 1,379 samples for PF and 1,460 samples for
non-forest (Figure 5).
**3.3.2 Assessment with NFI**
We selected the 5th to 9th NFI data to validate the accuracy of the produced PF maps in the years
1998, 2003, 2008, 2013, and 2018. Specifically, we compared the proportion of the estimated PF area in
31 provinces/regions (excluding Hong Kong, Macao, and Taiwan) of China with the NFI statistics. As
the Chongqing data are included in the Sichuan data in the 5th NFI, only 30 provinces' records were
used. The slope of the reduced major axis regression line (Cohen et al., 2003) and Pearson's
product-moment correlations (*r*) were chosen as metrics to evaluate the correlation between the two
datasets. In addition, we utilized data from the 5th and the 9th NFI to generate the PF expansion
reference data, representing the PF expansion from 1998 to 2018. Similarly, we compared the
proportions of the estimated PF expansion area in 30 provinces/regions of China with the reference data
of PF expansion.
**3.3.3 Comparison with existing products**
We compared the C-PFM-based PF map in this study with four existing PF products: Cheng's
product (Cheng et al., 2024), GFC2020 (Bourgoin et al., 2025), Du's planting year map (Du et al., 2022;
Harris et al., 2019), and SDPT_V2 (Richter et al., 2024). For consistency with the definition of NF, we
merged the naturally regenerating and primary forest classes in GFC2020 into a single NF category.
Additionally, to ensure the reliability and consistency of the comparison, all products were resampled to
a spatial resolution of 1 km. The comparison was performed using two evaluation strategies. First, we
assessed the accuracy of each product by validating it against visually interpreted reference samples
representing ground truth for the year 2020. In this strategy, the accuracy assessment was conducted
using a binary classification (PF and non-PF), as Du's planting year map and SDPT_V2 specifically
represent PF only. Second, we performed a time-series comparison over three periods (2010, 2015, and



2020), evaluating maps using three classes—PF, NF, and non-forest—based on validation samples
representing ground truth for the years 2010, 2015, and 2020. Due to the difficulty in collecting reliable
reference samples before 2010 using Google Earth imagery, earlier years were not included. This
approach was adopted because, among the four existing products, only Cheng's map provides
time-series classifications for both PF and NF.
**4 Results**
**4.1 Accuracy of the produced map**
As shown in Table 2, the 2020 map achieves an overall accuracy (OA) of 90.8% when validated
against the visually interpreted reference samples. This result indicates that the accuracy of the 2020
map is satisfactory and further demonstrates the acceptable performance of the proposed C-PFM
method. When validated using the additional time-series validation samples (Table 3), all periods
achieved an OA of above 80.0%, except 2014, 2016, and 2017, which still attained OAs exceeding
329    71.0%.

Table 2. Accuracy assessment (based on the methodology recommended by Olofsson et al (2014)) of NF and PF maps
for the year 2020 across China.

| Metrics | PF | NF | non-forest |
|---|---|---|---|
| UA (%) | 68.7 | 75.3 | 96.9 |
| PA (%) | 61.2 | 80.4 | 97.1 |
| OA (%) | | 90.8 | |

Table 3. Accuracy assessment against additional validation samples for different periods.

| Year | PF | | | NF | | | Non-forest | | | No. of |
|---|---|---|---|---|---|---|---|---|---|---|
| | OA | F1 | UA | PA | F1 | UA | PA | F1 | UA | PA | samples |
| | (%) | (%) | (%) | (%) | (%) | (%) | (%) | (%) | (%) | (%) | |
| 2000-2006 | 82.0 | 54.5 | 47.4 | 64.3 | 79.3 | 88.5 | 71.9 | 94.3 | 93.2 | 95.3 | 89 |
| 2007-2010 | 82.5 | 65.5 | 67.9 | 63.3 | 66.7 | 66.7 | 66.7 | 95.3 | 93.8 | 96.8 | 114 |
| 2011-2012 | 88.1 | 68.0 | 63.0 | 73.9 | 87.6 | 88.5 | 86.8 | 96.2 | 98.4 | 94.0 | 143 |
| 2013 | 80.4 | 74.6 | 77.2 | 72.1 | 68.7 | 67.6 | 69.7 | 90.8 | 88.9 | 92.8 | 163 |
| 2014 | 72.9 | 66.2 | 59.7 | 74.2 | 61.9 | 70.3 | 55.3 | 87.0 | 90.5 | 83.8 | 177 |
| 2015 | 81.0 | 75.2 | 74.5 | 75.9 | 77.2 | 78.6 | 75.9 | 88.1 | 88.1 | 88.1 | 142 |
| 2016 | 73.3 | 69.7 | 66.0 | 73.8 | 38.1 | 44.4 | 33.3 | 84.8 | 86.7 | 83.0 | 101 |
| 2017 | 71.7 | 69.6 | 67.1 | 72.3 | 64.6 | 75.0 | 56.8 | 80.0 | 76.6 | 83.7 | 145 |
| 2018 | 82.4 | 78.4 | 76.3 | 80.6 | 73.8 | 79.5 | 68.9 | 91.7 | 90.4 | 93.0 | 188 |
| 2019 | 80.5 | 74.3 | 74.7 | 73.9 | 52.4 | 52.4 | 52.4 | 96.0 | 95.5 | 96.4 | 241 |
| 2020 | 84.3 | 79.2 | 78.9 | 79.5 | 72.2 | 76.5 | 68.4 | 94.7 | 92.5 | 97.1 | 337 |




Figure 6. Annual PF and NF maps across China at a 30 m spatial resolution from 1990 to 2020 (12 years of maps are

shown).



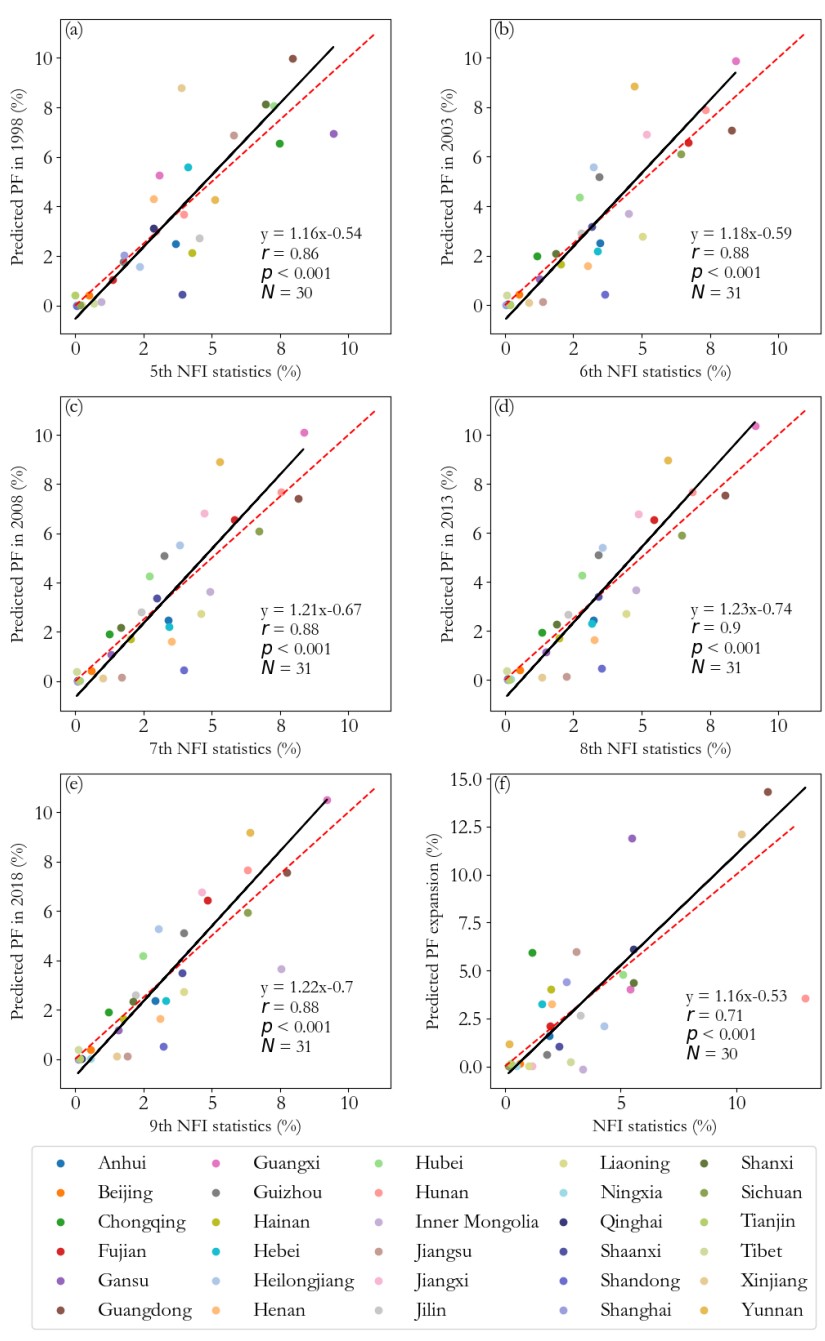


Figure 7. Comparisons between PF area estimates across provinces/regions of China and NFI statistics in (a) 1998, (b) 2003, (c) 2008, (d) 2013, and (e) 2018. (f) Comparison between PF expansion estimates from 1998 to 2018 and NFI statistics. The regression lines were derived using the reduced major axis analysis and the dashed lines represent the 1:1 line.





Figure 6 shows the generated annual maps in 12 selected years based on the C-PFM method. To
further assess the accuracy of the generated annual PF maps, we compared the proportions of estimated
PF in five years (i.e., 1998, 2003, 2008, 2013, and 2018) and PF expansion from 1998 to 2018 in China
with corresponding NFI statistics. As depicted in Figure 7(a)-(e), all the maps exhibit high correlations
with NFI statistics, with the lowest $r$ value of 0.85 for the year 1998 and the highest $r$ value of 0.91 for
the year 2013. Regarding PF expansion from 1998 to 2018 (Figure 7(f)), an $r$ value of 0.71 is observed.

### 347    4.2 Different training sample generation strategies

Three strategies for generating training samples across China were compared: (1) Extracting samples
from high-confidence areas identified by integrating multiple PF and NF maps in 2020; (2) Using only
the FD layer, as adopted in the study of Xiao et al. (2024); and (3) A hybrid approach. Specifically, for
the hybrid approach, PF samples were drawn from the high-confidence PF regions, while NF samples
were derived from the FD layer. This strategy is based on the observation that PF samples extracted
from multiple PF maps are more reliable than those derived from FD layers. Conversely, NF samples
extracted from multiple NF maps resulted in poorer model performance compared to those derived from
the FD layer. This discrepancy is likely due to the low consistency between the existing maps.
As shown in Table 4, the hybrid strategy yielded the most reliable samples, resulting in the highest
OA. Under this strategy, the F1-scores for all three classes (NF, PF, and non-forest) were the highest
among the three approaches. In contrast, the strategy based solely on high-confidence areas from PF and
NF maps produced the lowest F1-score and PA for the NF class. The superior performance of the hybrid
strategy reveals the more reliable NF samples derived from the FD layer and the PF samples extracted
from high-confidence PF regions across multiple products.
Table 4. Comparison of classification performance using different training sample generation strategies.

| Classes | Metrics | High-confidence areas | FD layer | Hybrid approach |
|---|---|---|---|---|
| | OA (%) | 79.9 | 75.9 | 82.3 |
| | F1-score (%) | 73.2 | 59.0 | 75.7 |
| PF | UA (%) | 74.5 | 85.5 | 77.9 |
| | PA (%) | 72.0 | 45.0 | 73.6 |
| | F1-score | 67.9 | 69.9 | 73.7 |
| NF | UA (%) | 67.6 | 56.1 | 72.0 |
| | PA (%) | 68.2 | 92.8 | 75.5 |
| | F1-score (%) | 92.6 | 91.6 | 93.0 |
| Non-forest | UA (%) | 91.4 | 89.1 | 92.0 |
| | PA (%) | 93.8 | 94.3 | 94.0 |






### 4.3 Influence of parameter settings in the C-PFM model

A series of experiments were conducted to determine the optimal parameters for the C-PFM model, including training sample size and feature selection. Specifically, an experiment was carried out to identify the optimal number of training samples per class within each tile. As shown in Figure 8, the OA stabilizes once the sample size exceeds 200 per class. The highest accuracy is achieved with 300 samples per class, resulting in an OA of 82.4% and an F1-score of 76.1% for PF.

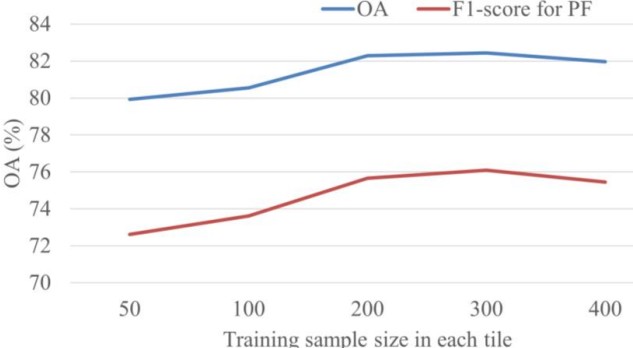

Figure 8. OA and F1-score for PF with varying training sample sizes.

Figure 9 presents the classification accuracy across seven climatic zones and the entire area (Entire) under different input feature sets. From the figure, we observe that using all features yields the highest accuracy in the entire area. This improvement is particularly pronounced in the Northern Subtropical Zone (NSTZ), suggesting that a richer feature set enhances the model's ability to distinguish PF in heterogeneous environments. In contrast, the Tropical Zone (TZ) achieved the highest accuracy when the top 60 features were used, which may be attributed to the concentration of large-scale plantations in this region, such as those in Hainan Province. Meanwhile, the Middle Temperate Zone (MTZ) attained the highest accuracy with only the top 30 features, possibly due to relatively homogeneous afforestation activities associated with programs such as the Three-North Forest Shelterbelt Program.

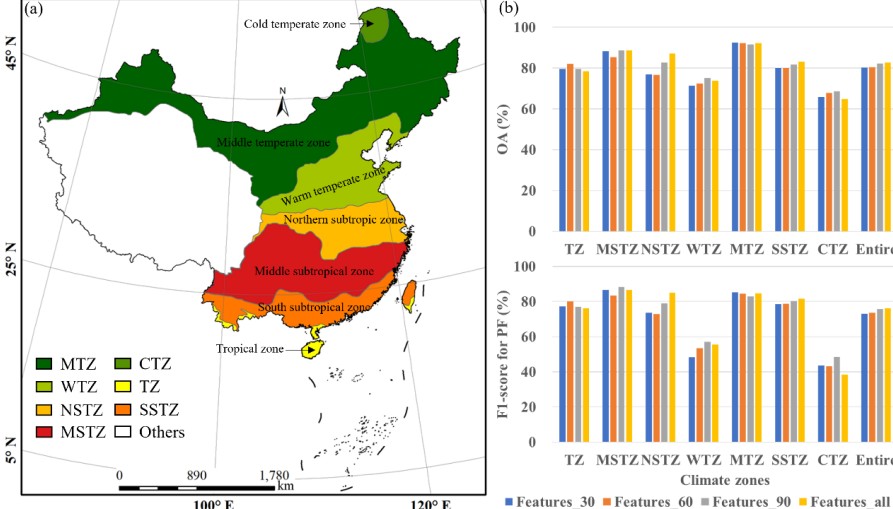


Figure 9. Classification performance under different input feature sets across seven climate zones and the entire area.
The feature sets include the top 30, 60, 90, and all 261 features ranked by importance. Climate zones are abbreviated as
follows: MTZ – middle temperate zone; CTZ – cold temperate zone; WTZ – warm temperate zone; TZ – tropical zone;
NSTZ – northern subtropical zone; SSTZ – southern subtropical zone; MSTZ – middle subtropical zone. (a) The
climate zones of China; (b) OA and F1-score for PF.
**4.4 Comparison with other products**

The spatial distribution of PF in this study and four existing PF products is in Figure 10. The

accuracy evaluation is shown in Figure 11 and Figure 12. It is seen from Figure 11 that the proposed
C-PFM achieves a higher OA of 69.9% and a comparable F1-score of 0.559. When validated using
time-series reference samples, Cheng's map yielded OAs ranging from 50.6% to 66.7% and PF
F1-scores between 0.353 and 0.582, as shown in Figure 12. Moreover, the C-PFM achieved higher OAs
ranging from 61.6% to 71.6%, with PF F1-scores between 0.537 and 0.563.

Figure 10. The PF maps of the C-PFM-based method and four existing products. (a) Cheng's product; (b) GFC2020; (c)

Du's product; (d) SDPT_V2; (e) C-PFM-based map.



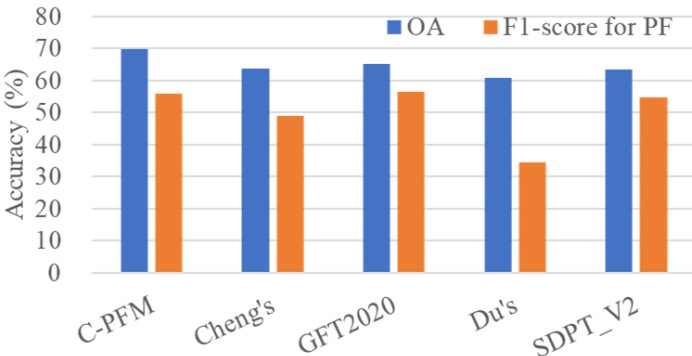


Figure 11. Comparison of PF mapping accuracy between the proposed C-PFM method and four existing PF products
for the year 2020.

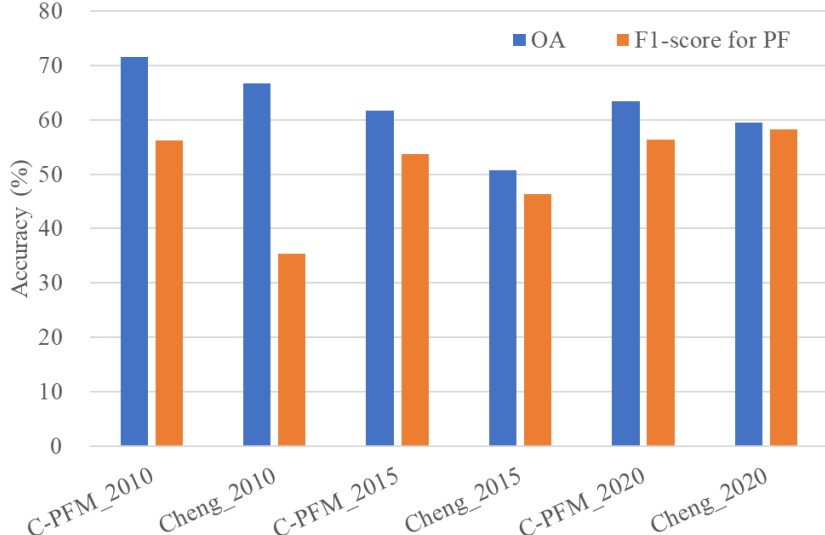


Figure 12. Comparison of PF mapping accuracy between the proposed C-PFM method and Cheng's product for
multiple years.
**4.5 Forest dynamic changes at national and provincial scales**
The trends of PF and NF in China from 1990 to 2020 are shown in Figure 13. For NF, the area
experienced a brief increase from 1990 to 1992, after which it began to decline. In contrast, PF showed
a dramatic increase from 1990 to 1995, followed by a stable increase until 2020. Table 5 shows that
China experienced a net forest gain of 8.06 Mha over the study period. Specifically, the area of PF
increased from 73.64 Mha in 1990 to 89.79 Mha in 2020, representing a net gain of 16.15 Mha. In
contrast, the area of NF declined from 158.95 Mha to 150.86 Mha, representing a net loss of 8.09 Mha.





Most provinces exhibited net forest expansion, with particularly notable increases in Sichuan, Shaanxi,
Inner Mongolia, and Hebei.

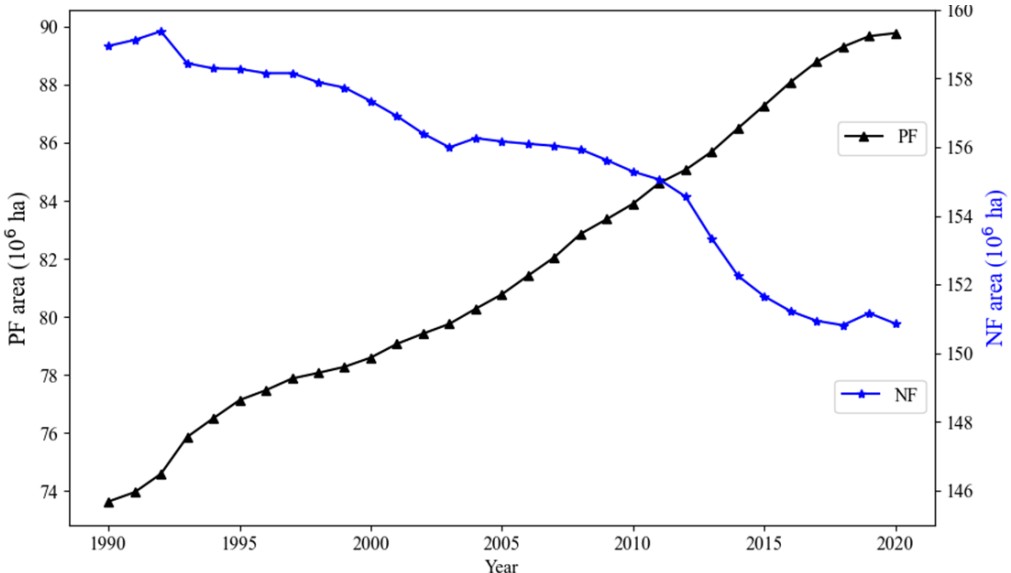


Figure 13. Dynamics of area change for PF and NF in the entire China from 1990 to 2020.

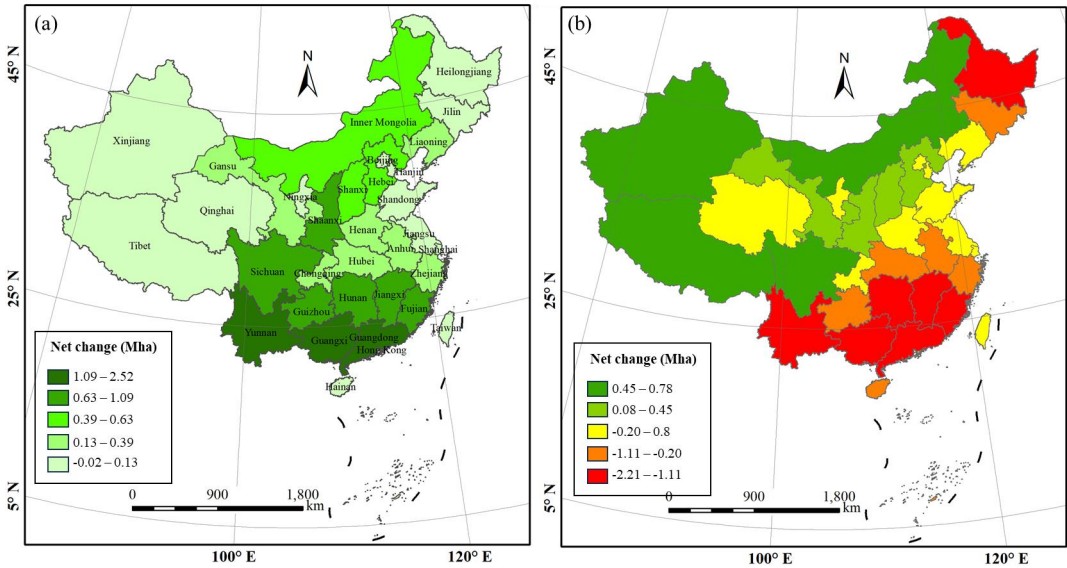


Figure 14. Net change of forest areas in the 34 provinces from 1990 to 2020. (a) PF. (b) NF.




Table 5. Net change of NF and PF area in each province from 1990 to 2020.

| Province | PF | | | NF | | | Forest (PF + NF) | | |
|---|---|---|---|---|---|---|---|---|---|
| | 1990 (Mha) | 2020 (Mha) | Net change (Mha) | 1990 (Mha) | 2020 (Mha) | Net change (Mha) | 1990 (Mha) | 2020 (Mha) | Net change (Mha) |
| Anhui | 1.83 | 2.08 | 0.25 | 1.78 | 1.58 | -0.20 | 3.61 | 3.66 | 0.05 |
| Beijing | 0.32 | 0.35 | 0.03 | 0.42 | 0.45 | 0.03 | 0.74 | 0.80 | 0.06 |
| Chongqing | 1.37 | 1.76 | 0.39 | 2.62 | 2.69 | 0.07 | 3.99 | 4.45 | 0.46 |
| Fujian | 4.58 | 5.67 | 1.09 | 5.59 | 4.30 | -1.29 | 10.16 | 9.97 | -0.20 |
| Gansu | 0.74 | 1.11 | 0.36 | 2.49 | 2.83 | 0.33 | 3.24 | 3.94 | 0.70 |
| Guangdong | 5.06 | 6.63 | 1.56 | 6.28 | 4.74 | -1.53 | 11.34 | 11.37 | 0.03 |
| Guangxi | 6.75 | 9.27 | 2.52 | 9.77 | 7.56 | -2.21 | 16.51 | 16.83 | 0.32 |
| Guizhou | 3.82 | 4.61 | 0.79 | 6.60 | 6.19 | -0.41 | 10.43 | 10.80 | 0.38 |
| Hainan | 1.29 | 1.40 | 0.11 | 1.01 | 0.81 | -0.20 | 2.29 | 2.21 | -0.08 |
| Hebei | 1.51 | 2.10 | 0.59 | 2.04 | 2.40 | 0.36 | 3.55 | 4.50 | 0.95 |
| Heilongjiang | 4.56 | 4.67 | 0.11 | 18.82 | 17.50 | -1.32 | 23.37 | 22.17 | -1.20 |
| Henan | 1.17 | 1.44 | 0.27 | 1.60 | 1.51 | -0.09 | 2.78 | 2.96 | 0.18 |
| Hong Kong | 0.03 | 0.03 | 0.00 | 0.05 | 0.04 | -0.01 | 0.08 | 0.07 | -0.01 |
| Hubei | 3.30 | 3.68 | 0.39 | 5.76 | 5.34 | -0.42 | 9.06 | 9.03 | -0.03 |
| Hunan | 5.72 | 6.77 | 1.05 | 7.63 | 6.32 | -1.31 | 13.35 | 13.08 | -0.26 |
| Inner Mongolia | 2.75 | 3.26 | 0.50 | 14.22 | 14.85 | 0.63 | 16.97 | 18.11 | 1.14 |
| Jiangsu | 0.13 | 0.11 | -0.02 | 0.09 | 0.05 | -0.03 | 0.22 | 0.16 | -0.05 |
| Jiangxi | 4.88 | 5.94 | 1.06 | 6.02 | 4.65 | -1.37 | 10.90 | 10.59 | -0.31 |
| Jilin | 2.28 | 2.32 | 0.04 | 6.25 | 5.91 | -0.33 | 8.53 | 8.23 | -0.29 |
| Liaoning | 2.02 | 2.40 | 0.38 | 2.81 | 2.74 | -0.07 | 4.83 | 5.14 | 0.31 |
| Macao | 0.00 | 0.00 | 0.00 | 0.00 | 0.00 | 0.00 | 0.00 | 0.00 | 0.00 |
| Ningxia | 0.02 | 0.03 | 0.01 | 0.04 | 0.06 | 0.02 | 0.06 | 0.09 | 0.03 |
| Qinghai | 0.02 | 0.03 | 0.01 | 0.52 | 0.59 | 0.08 | 0.54 | 0.62 | 0.08 |
| Shaanxi | 2.24 | 3.11 | 0.87 | 5.78 | 6.23 | 0.45 | 8.02 | 9.34 | 1.32 |
| Shandong | 0.33 | 0.46 | 0.13 | 0.32 | 0.32 | 0.01 | 0.64 | 0.78 | 0.14 |
| Shanghai | 0.00 | 0.00 | 0.00 | 0.00 | 0.00 | 0.00 | 0.00 | 0.00 | 0.00 |
| Shanxi | 1.45 | 2.09 | 0.63 | 1.72 | 1.96 | 0.23 | 3.18 | 4.04 | 0.87 |
| Sichuan | 4.56 | 5.34 | 0.78 | 13.49 | 14.27 | 0.78 | 18.05 | 19.61 | 1.57 |
| Taiwan | 1.02 | 1.07 | 0.05 | 1.50 | 1.41 | -0.09 | 2.52 | 2.49 | -0.04 |
| Tianjin | 0.02 | 0.02 | 0.01 | 0.02 | 0.01 | 0.00 | 0.03 | 0.03 | 0.00 |
| Tibet | 0.31 | 0.35 | 0.05 | 9.78 | 10.44 | 0.66 | 10.09 | 10.80 | 0.71 |
| Xinjiang | 0.05 | 0.09 | 0.04 | 0.99 | 1.73 | 0.74 | 1.04 | 1.82 | 0.78 |
| Yunnan | 6.33 | 8.17 | 1.84 | 19.21 | 18.11 | -1.11 | 25.54 | 26.28 | 0.73 |
| Zhejiang | 3.18 | 3.42 | 0.24 | 3.76 | 3.27 | -0.49 | 6.93 | 6.69 | -0.25 |
| China | 73.64 | 89.79 | 16.15 | 158.95 | 150.86 | -8.09 | 232.59 | 240.65 | 8.06 |


As shown in Figure 14, most provinces experienced a net increase in PF. The top five provinces with
the largest net increases in PF are Guangxi, Yunnan, Guangdong, Fujian, and Jiangxi, each contributing
more than 1.06 Mha. These provinces played a key role in driving the overall forest expansion in China.





In contrast, NF declined in most provinces. The greatest net decreases in NF were observed in Guangxi,
Guangdong, Jiangxi, Heilongjiang, and Hunan. However, a few provinces exhibited net gains in NF,
including Sichuan (0.78 Mha), Xinjiang (0.74 Mha), Tibet (0.66 Mha), Inner Mongolia (0.63 Mha),
Shaanxi (0.45 Mha), and Hebei (0.36 Mha), which recorded the highest increases in NF area.
In summary, forest expansion in China can be attributed to favorable climatic and geographical
conditions that support forest regrowth in the southern regions, as well as the implementation of
large-scale afforestation and reforestation programs in the north. Under the Natural Forest Protection
Program (Yan et al., 2022), several provinces, including Sichuan, Xinjiang, Tibet, Inner Mongolia,
Shaanxi, and Hebei, showed notable increases in NF area, highlighting the program's effectiveness in
these regions. In contrast, forest expansion in provinces such as Heilongjiang and Guangxi was limited,
likely due to continued timber supply demands in those areas.

### 4.6 Change pattern of PF and NF

Figure 15 illustrates the annual dynamics of the PF area across various provinces from 1990 to 2020.
It shows that Guangxi, Yunnan, and Guangdong had relatively large PF areas in 1990 and subsequently
experienced rapid increases, with slopes greater than 0.0587, indicating significant expansion trends.
Guangxi exhibited the largest slope at 0.0818, and the polyline in Figure 15(a) reveals a dramatic
increase in PF area after 2004, likely due to the extensive planting of eucalyptus trees during this period.
In addition to these provinces, Fujian, Jiangxi, Shaanxi, and Hunan also display high growth trends,
indicating substantial planting efforts in those regions after 1990. In contrast, most other provinces
exhibit slow increases or stable PF areas, likely due to natural forest protection projects (Yan et al., 2022)
and limited planting conditions.
Regarding the dynamics of the NF area throughout the study period, most provinces experienced
stable or slight decreases in NF area (Figure 16). Specifically, Xinjiang, Tibet, Inner Mongolia, Sichuan,
and Hebei exhibit positive slopes, suggesting effective NF protection measures in these regions. In
contrast, Guangxi, Guangdong, Hunan, Yunnan, and Fujian show the lowest negative slopes, indicating
a tendency towards deforestation in these provinces.








Figure 15. Annual dynamics of PF area from 1990 to 2020 across various provinces.


Figure 16. Annual dynamics of NF area from 1990 to 2020 across various provinces.

### 4.7 Forest expansion in different forest ecological engineering areas

To further evaluate the drivers of forest regrowth in China, we calculated the area of PF and NF and
their growth rate from 1990 to 2020 within eight forestry ecological engineering areas (Figure 17).
These areas include the Three-North Forest Shelterbelt Program (TNSP), Taihang Mountain Greening





Project (TMGP), Liaohe River Shelterbelt Program (LHSP), Yellow River Shelterbelt Program (YRSP),
Huaihe River and Taihu lake Shelterbelt Program (HTSP), upper and middle reaches of Yangtze river
Shelterbelt Program (YZSP), Pearl River Shelterbelt Program (PRSP), and Coast Shelterbelt Program
(COSP) (Liu et al., 2023).
Figure 17 shows that all eight forestry ecological engineering areas present a positive growth rate of
PF (ranging from 8.20% to 53.90%). Notably, the YRSP and TMGP experienced the most substantial
PF growth, with increases of 53.90% and 37.40%, respectively. While most forest ecological
engineering areas experienced negative growth rates in NF, a few regions exhibited positive trends.
Specifically, the TNSP (24.80%), TMGP (16.30%), YRSP (15.90%), and LHSP (5.40%) showed
increases in NF area. The COSP exhibited the lowest growth rate at -23.20%, followed by the PRSP
(-19.10%) and HTSP (-17.70%). The low growth rates observed in the COSP and HTSP regions may be
attributed to the degradation of NF, likely due to intensified human activities in these areas. In contrast,
the decline in NF in the PRSP region may result from extensive PF expansion, as PF exhibited a growth
rate of 34.00%, likely encroaching on NF areas. Overall, forest expansion in these regions appears to be
primarily driven by the growth of PF.

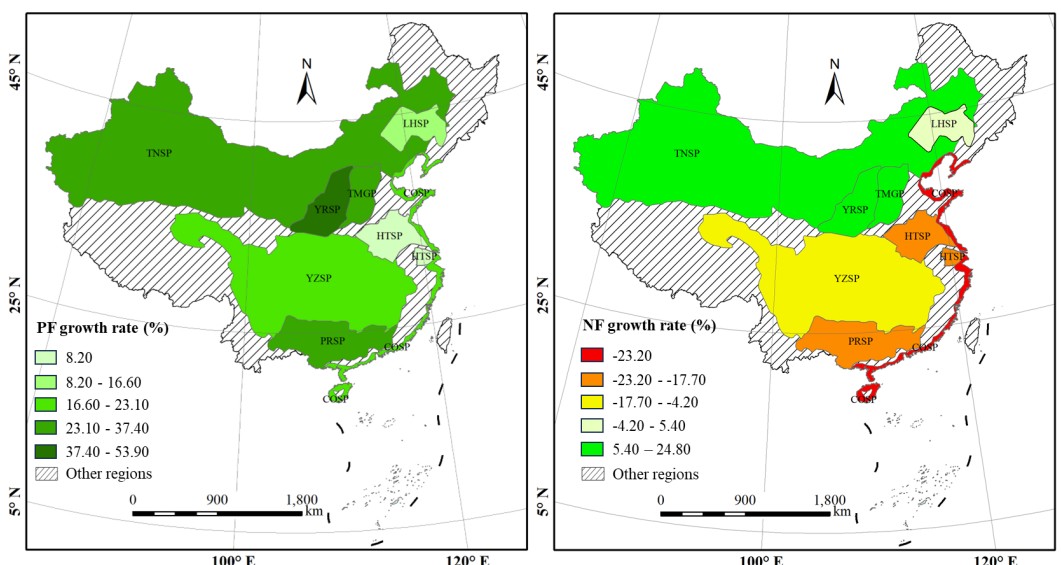

Figure 17. PF and NF area growth rate from 1990 to 2020 across various forest ecological engineering areas.
PF in all eight forestry ecological engineering areas exhibits an overall increase trend during the
study period (Figure 18 (a)). In contrast, NF in most of these regions showed either a decline or stable
trend, with the exception of the TNSP (Figure 18 (b)), where both PF and NF increased. Specifically,
the YZSP region experienced the most significant PF expansion, with a marked increase from 1990 to





2020. By contrast, the NF in YZSP exhibited a decline trend from 1990 to 2020, likely due to the
conversion of NF into PF in the region. In the PRSP, PF demonstrated a relatively stable and continuous
increase throughout the study period. In the TNSP, both PF and NF showed consistent growth over the
entire study period.

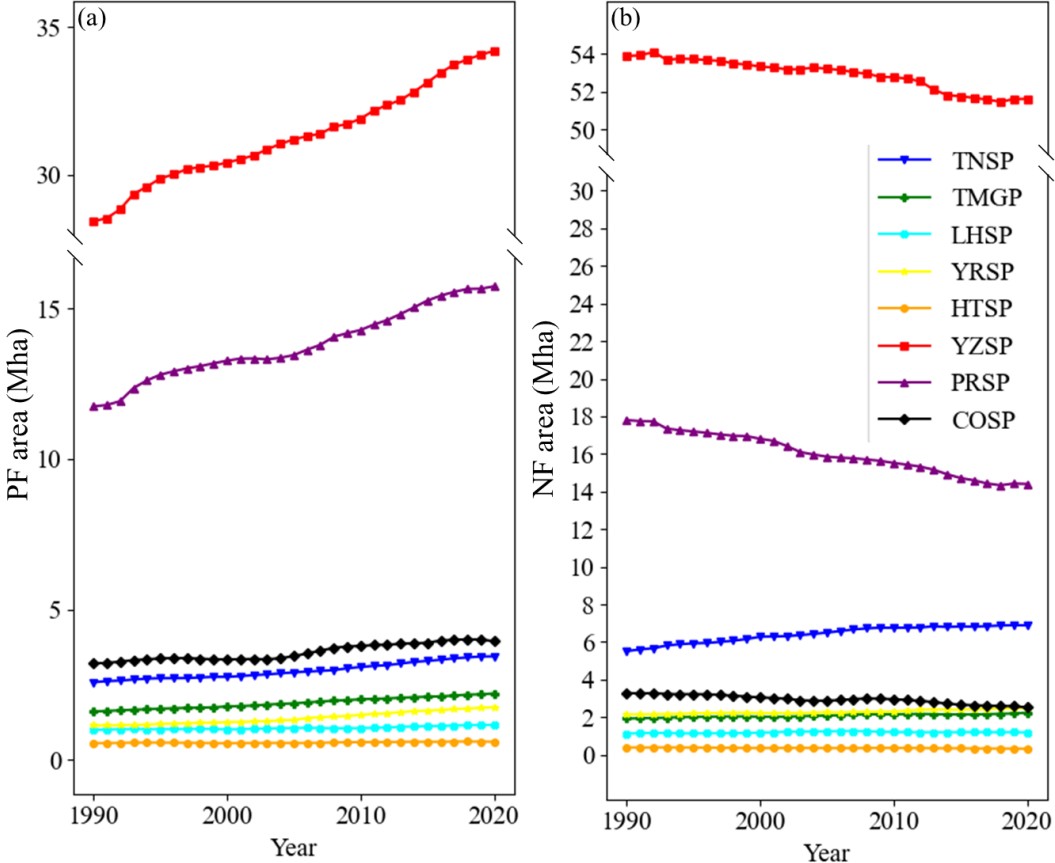


Figure 18. Annual dynamics of PF and NF area from 1990 to 2020 across various forest ecological engineering areas.
(a) PF. (b) NF.
**5 Discussion**
**5.1 The impact of different vegetation and regions**
Different vegetation types, particularly coniferous and deciduous forests, exhibit distinct structural
and phenological characteristics, which can influence their spectral and temporal signatures and
subsequently affect model performance. To address these differences, we evaluated the classification
performance by forest type, including evergreen needleleaf forest (ENF), evergreen broadleaf forest



(EBF), deciduous needleleaf forest (DNF), deciduous broadleaf forest (DBF), and mixed forest (MF).
Specifically, we extracted forest type information from the Terra and Aqua combined Moderate
Resolution Imaging Spectroradiometer (MODIS) Land Cover Type (MCD12Q1) Version 6.1 product
(Friedl and Sulla-Menashe, 2022), which provides global coverage at a spatial resolution of 500 m. Due
to the limited distribution of DNF in China, this class was excluded from the accuracy assessment. As
illustrated in Figure 19, the DBF class achieved the highest F1-score for PF, whereas the ENF class
yielded the lowest F1-score. This discrepancy may be attributed to the phenological variability and
canopy textural differences inherent in DBF, such as seasonal leaf shedding and more pronounced
structural heterogeneity, which provide richer information for distinguishing PF from NF.

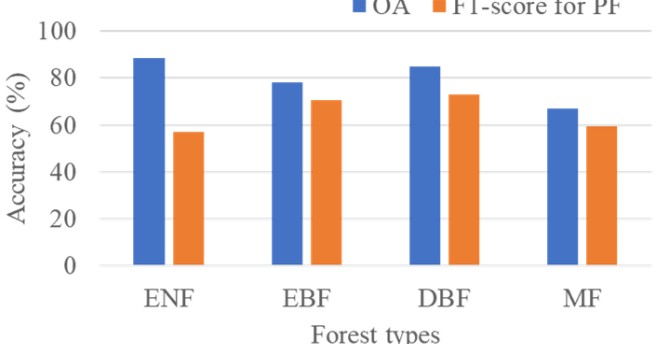


Figure 19. Classification performance of PF and NF across different forest types. ENF: evergreen needleleaf forest;
EBF: evergreen broadleaf forest; DBF: deciduous broadleaf forest; MF: mixed forest.
Given China's vast geographic expanse, regional variations in mapping accuracy are to be expected.
Thus, a spatially stratified accuracy assessment was conducted to examine classification performance
across different regions. A spatially explicit accuracy assessment was conducted based on the major
climate zones of China (Figure 9(a)). The results revealed substantial regional variation in classification
accuracy. As shown in Figure 20, the MTZ and MSTZ exhibited the highest OA and F1-score for PF
classification, respectively, followed by SSTZ and TZ. In contrast, the CTZ demonstrated the lowest
classification performance. The relatively high accuracy observed in the tropical and subtropical zones
may be attributed to the concentration of large-scale plantations in these regions, which benefit from
favorable temperature and precipitation conditions. Conversely, the lower accuracy in the CTZ could be
due to persistent snow cover and the dominance of coniferous forests, whose homogeneous canopy
texture complicates the differentiation between PF and NF. These findings underscore the influence of
climatic heterogeneity as a significant source of uncertainty in large-scale forest mapping.



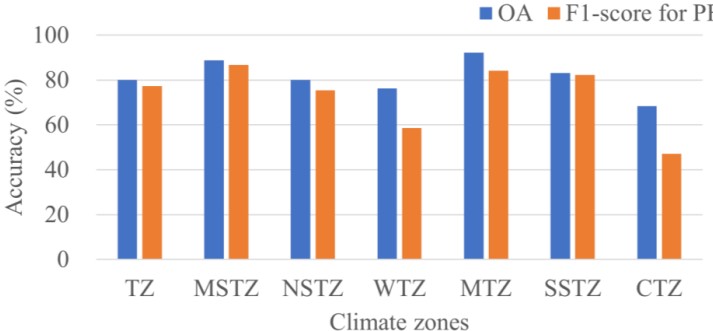

Figure 20. Classification performance of PF and NF across different climate zones of China. MTZ: middle temperate zone; CTZ: cold temperate zone; WTZ: warm temperate zone; TZ: tropical zone; NSTZ: northern subtropical zone; SSTZ: southern subtropical zone; MSTZ: middle subtropical zone.

Additionally, accuracy was evaluated within the seven natural regions of China (i.e., Northeast, North, East, South, Central, Southwest, and Northwest). As shown in Figure 21, higher accuracy was observed in the South region (i.e., Hainan and Guangxi provinces), where forest types are more homogeneous. In contrast, lower accuracy was noted in the East region, where mixed forest types introduce considerable spectral variability. In these areas, small patches of plantation forests often contain spectrally mixed pixels, resulting in increased omission and commission errors.

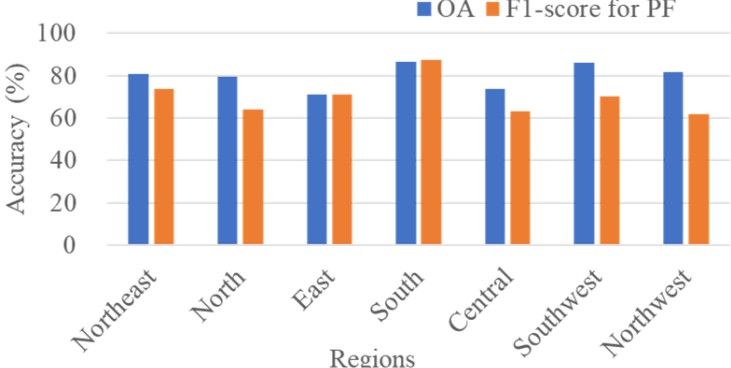

Figure 21. Classification performance of PF and NF mapping across the seven natural regions of China.

## 5.2 Analysis of model fitting ability and generalization

A series of supplementary analyses were conducted across 16 testing regions (with locations listed in Table 6) to evaluate three key aspects of the model uncertainty: (1) fitting ability, (2) spatial generalization across regions, and (3) sensitivity to key parameters of the RF classifier.





Table 6. Location information on the 16 testing regions.

| Regions | Minimum longitudes | Minimum latitudes | Maximum longitudes | Maximum latitudes |
|---|---|---|---|---|
| Region 1 | 102.5 | 35.0 | 103.0 | 35.5 |
| Region 2 | 104.0 | 27.0 | 104.5 | 27.5 |
| Region 3 | 105.5 | 34.0 | 106.0 | 34.5 |
| Region 4 | 108.5 | 23.0 | 109.0 | 23.5 |
| Region 5 | 109.5 | 19.0 | 110.0 | 19.5 |
| Region 6 | 110.5 | 33.0 | 111.0 | 33.5 |
| Region 7 | 112.0 | 36.0 | 112.5 | 36.5 |
| Region 8 | 114.0 | 23.5 | 114.5 | 24.0 |
| Region 9 | 114.0 | 29.5 | 114.5 | 30.0 |
| Region 10 | 116.0 | 27.5 | 116.5 | 28.0 |
| Region 11 | 117.0 | 41.5 | 117.5 | 42.0 |
| Region 12 | 118.0 | 44.0 | 118.5 | 44.5 |
| Region 13 | 121.0 | 41.5 | 121.5 | 42.0 |
| Region 14 | 122.5 | 50.5 | 123.0 | 51.0 |
| Region 15 | 123.5 | 53.0 | 124.0 | 53.5 |
| Region 16 | 126.0 | 50.5 | 126.5 | 51.0 |

First, fitting ability was evaluated. Specifically, a stratified cross-validation analysis was conducted
by dividing the dataset in each testing region into training (80%) and validation (20%) subsets, which
were then used to train and evaluate the RF classifier. The results revealed a relatively small
performance gap between training and validation accuracies. Specifically, in 10 out of the 16 testing
regions, the overfitting gap (defined as the difference between training and validation accuracy) was
below 20% (Figure 22). This indicates that the model exhibits reliable generalization capability and
does not suffer from obvious overfitting in the majority of regions.



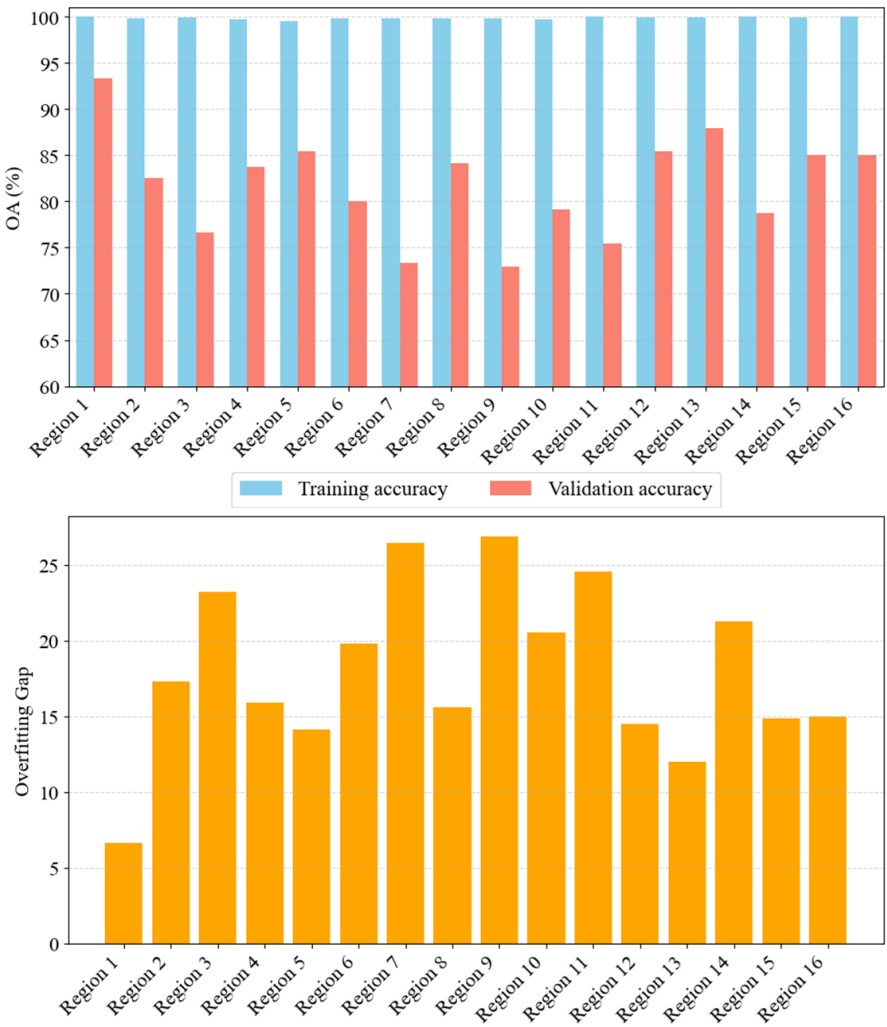

Figure 22. Training and validation accuracies of the RF classifier across the 16 testing regions. The overfitting gap is defined as the difference between training and validation accuracy.

Second, the model's spatial transferability (generalization performance) in geographically independent regions was assessed. Specifically, a leave-one-block-out cross-validation approach (Fu et al., 2004), in which each region was retained for validation while the model was trained on samples from the other regions, was implemented. The results show that while the model's performance remained relatively stable, it exhibited lower validation accuracies (ranging from 62.3% to 73.8%) and larger overfitting gaps across different testing regions (Figure 23). These findings suggest limited generalization capacity across regions. The reason is that a local RF classifier was used in the paper, which was trained and predicted for each tile separately. This scheme, however, is more appropriate for

large-scale mapping across China, where regional ecological and spectral heterogeneity is substantial.

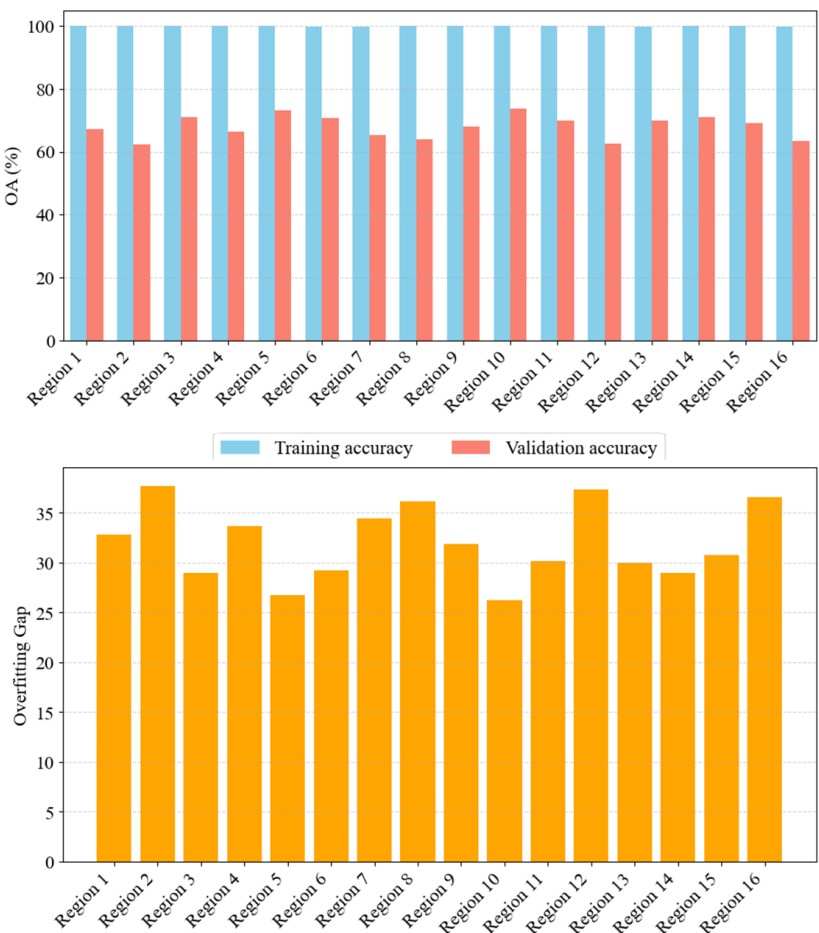


Figure 23. Spatial generalization performance based on leave-one-block-out cross-validation across the 16 testing
regions.

Third, the sensitivity of the model to a key parameter of the RF classifier—namely, the number of

decision trees (*n_tree*)—across the testing regions was examined. As shown in Figure 24, classification
performance remained relatively stable once the number of trees exceeded 100, indicating that the
model is robust to variations in this parameter.

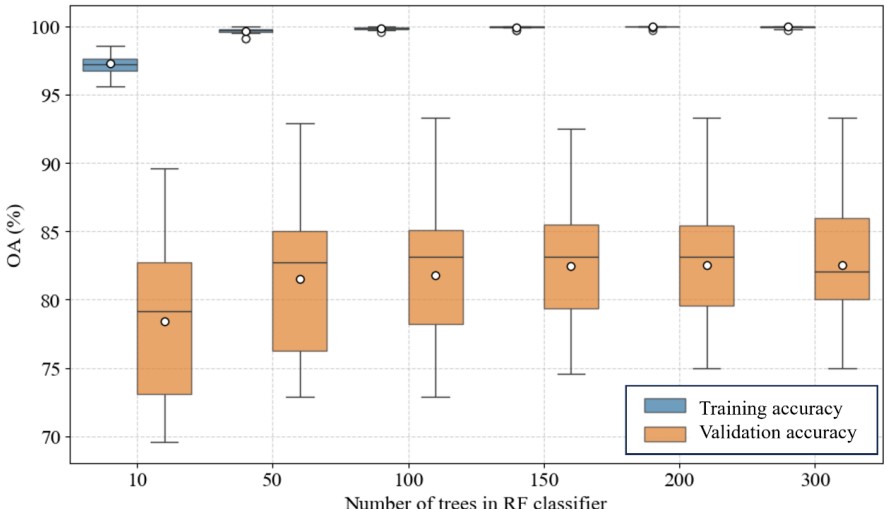


Figure 24. Classification performance of the RF classifier with varying number of decision trees (*n_tree*).

**5.3 The driving forces behind forest expansion**
China's strategic implementation of the Natural Forest Protection Program was a direct response to
severe ecological degradation resulting from decades of overexploitation of NF resources, particularly
in the aftermath of the catastrophic floods in 1998. Our results show a marked increase in the PF area in
the PRSP region before 2000, followed by a noticeable slowdown in growth thereafter. It is a trend
likely attributable to the introduction of the Natural Forest Protection Program. Conversely, in the TNSP
region, the area of NF began to show a steady and sustained increase after this period, indicating the
effectiveness of NF protection and restoration policies. Moreover, the continued expansion of PF across
all eight major forestry ecological engineering regions highlights the success of large-scale afforestation
and reforestation programs implemented to address environmental degradation and timber supply
shortages. Notable initiatives include the Natural Forest Protection Program, the Three-North
Shelterbelt Project, and China's Conversion of Cropland to Forest Program (Guti érrez Rodr íguez et al.,
2016), all of which have played pivotal roles in promoting forest recovery at the national scale. In
addition to policy-driven interventions, favorable climatic and geographic conditions in southern China,
such as warm temperatures and abundant precipitation, have further facilitated PF expansion. Provinces
like Guangxi, Guangdong, and Yunnan, which benefit from both optimal biophysical conditions and
strong policy support, have experienced the most substantial gains in PF area.
Conversely, the decline of NF in many regions is closely associated with anthropogenic pressures and
land-use changes. Although the Natural Forest Protection Program was designed to curb logging and



safeguard natural ecosystems, the conversion of NF to PF continued in certain areas, particularly during the early years of policy implementation. This trend was largely driven by the economic incentives associated with PF, which typically offers faster returns and clearer land tenure arrangements. For instance, in the YZSP region, a substantial decline in NF was observed between 1990 and 2020, accompanied by a rapid expansion of PF, which highlights the trade-off between ecological conservation and economic development.

Moreover, persistent timber demand in provinces such as Heilongjiang and Guangxi has contributed to the ongoing NF loss (Zhang et al., 2014), despite national efforts to enforce forest protection regulations. These cases underscore the challenges of balancing ecological goals with local socio-economic needs and emphasize the importance of strengthening policy enforcement, promoting sustainable forest management, and ensuring that afforestation initiatives do not inadvertently lead to the degradation of natural forest ecosystems.

While the expansion of PF has contributed to increased forest cover and carbon sequestration, it also raises considerable ecological concerns (Xu et al., 2023). In contrast to NF, which harbors rich biodiversity, complex structural layers, and robust ecosystem functions, PF (particularly those dominated by monocultures) typically offer lower habitat heterogeneity, reduced resistance to pests and diseases, and heightened vulnerability to climate variability. The large-scale replacement of NF with PF may therefore lead to declines in native biodiversity, degradation of ecosystem services, and increased ecological fragility, particularly in ecologically sensitive and biodiversity-rich regions. Moreover, the intensive management regimes associated with PF, which are characterized by short rotation cycles and recurrent disturbances, can result in soil nutrient depletion, hydrological disruption, and long-term declines in forest health and productivity if not appropriately regulated. The widespread use of fast-growing, often non-native species in afforestation initiatives can further aggravate ecological imbalances by suppressing native flora and altering local ecosystem dynamics. These risks underscore the need for a more ecologically informed approach to PF management that balances production goals with long-term ecosystem sustainability.

### 5.4 The advantages of the proposed C-PFM method

The proposed C-PFM method offers a novel approach to monitoring PF expansion throughout the Landsat records. The C-PFM method holds several advantages over traditional methods. It leverages the forest disturbance detection algorithms (i.e., CCDC) to identify PF expansion areas, providing a new strategy for monitoring forest expansion. Traditional methods typically rely on multi-temporal maps,



which often exhibit inconsistent quality due to the variation in training samples across different periods.
Some studies, such as Cheng et al., (2024), utilized samples from undisturbed areas in disturbance
analysis algorithms. However, this approach may introduce issues of sample imbalance, as it lacks
sufficient representation from disturbed areas. Alternatively, the C-PFM auto-generates numerous
training samples, which is essential for large-scale supervised classification but challenging to obtain
through traditional methods. Manual collection of training samples is time-consuming and
labor-intensive, particularly at national or global scales. Our approach involves integrating training
samples from time-series analysis and existing NF and PF datasets. Furthermore, the C-PFM based on
the GEE cloud platform enables large-scale, fine spatial resolution mapping, which is crucial for
accurately assessing forest regrowth in China's diverse landscapes.
The annual NF and PF datasets produced by the C-PFM method are the first to describe yearly
dynamical changes at a 30 m resolution across China. Existing relative datasets, such as the one
produced by Cheng et al. (2024), provide data with a 5-year interval. The finer temporal resolution of
the dataset produced in this paper is more beneficial for monitoring forest dynamics, especially the
dramatic expansion of PF in China. Additionally, the fine spatial resolution of 30 m is suitable for
monitoring areas with smaller-scale forests (Xiao et al., 2023).

### 5.5 Potential applications

Over recent decades, China has led global greening efforts through extensive afforestation,
reforestation, and forest conservation projects (Wang et al., 2007; Qiu et al., 2017), establishing itself as
a leader in these initiatives (Chen et al., 2019). However, the expansion of PF remains controversial due
to its potential impacts on ecosystems, biodiversity, and native tree species (Fagan et al., 2022).
Accurate monitoring of NF and PF expansion is crucial for evaluating carbon sequestration estimates
and assessing the progress of forestry ecological engineering initiatives, especially as China is
navigating the dual challenges of ensuring food security and pursuing carbon neutrality. The proposed
C-PFM method offers a valuable tool for mapping the PF expansion, contributing to environmental
improvements and the evaluation of China's afforestation efforts.
Our dataset is a crucial resource for environmental researchers and policymakers, aiding in the
development of more effective and ecologically sound afforestation strategies. There is a need for
significant improvements in the survival rates of PF and the ecosystem services they provide (Cao et al.,
2011). Xu (2011) highlighted that some afforestation efforts in China involve planting trees in areas
where they did not historically grow, which may not be the most effective approach for environmental





enhancement. Moreover, monoculture planting practices can reduce resilience to disturbances and offer
fewer ecosystem services compared to natural forests (Lian et al., 2023; Betts et al., 2022). Planting the
right species in appropriate locations is essential for successful afforestation (Xu et al., 2023). Thus, our
dataset, which details PF and NF expansion, is vital for improving the quality of afforestation practices
and ensuring more beneficial environmental outcomes.

**5.6 Uncertainty and future work**

A pixel-level uncertainty analysis for the PF and NF maps for the year 2020 was conducted.
Specifically, the C-PFM-based PF maps were compared simultaneously with four existing PF products:
Cheng's product, GFC2020 (Bourgoin et al., 2025), Du's planting year map (Harris et al., 2019; Du et
al., 2022), and SDPT_V2 (Richter et al., 2024). A score (ranging from 1 to 4) representing the
agreement between the maps was recorded for each pixel. This approach is particularly effective in
situations where pixel-level reference data is limited or unavailable at the national scale. The spatial
agreement map shown in Figure 25 presents an agreement count, indicating the number of existing PF
products that are consistent with the C-PFM-based map in identifying PF at each location. This
agreement counts as a proxy for the uncertainty map. The uncertainty map revealed that 72.8% of the
identified PF pixels in this study correspond to areas that agree with at least one of the existing products
(Figure 26). The remaining 27.2% of PF pixels were uniquely identified in this study, which can be
partly attributed to the finer spatial resolution employed compared to other products. Additionally, we
observed that provinces such as Hainan, Fujian, Guangxi, and Guangdong exhibited relatively low
uncertainties (i.e., a high proportion of agreement in PF identification), whereas regions like Xinjiang,
Qinghai, Shanghai, and Tibet showed relatively higher uncertainties (i.e., lower PF agreement ratios).
This suggests that the C-PFM method performs more reliably in areas with extensive PF coverage,
particularly in tropical and subtropical regions, where environmental conditions such as temperature and
precipitation are more favorable for tree growth.



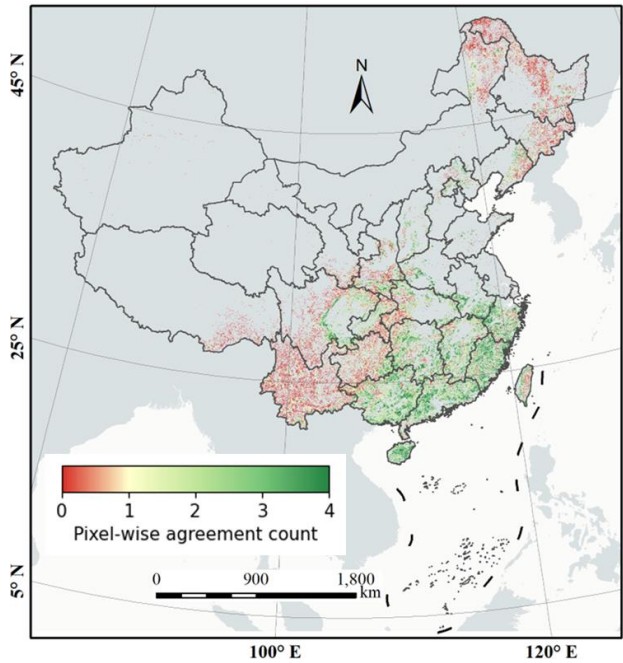


Figure 25. Pixel-level uncertainty map showing the agreement between the C-PFM-derived PF map for 2020 and four
existing PF products. The map reflects the number of products that are consistent with the C-PFM classification at each
pixel, serving as a proxy for spatial uncertainty.

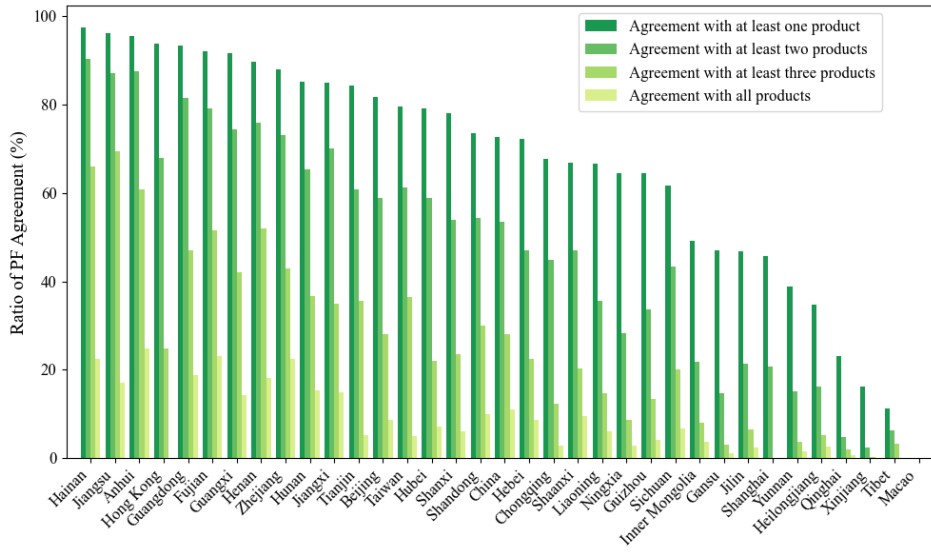


Figure 26. Proportion of the PF pixels at different confidence levels. The confidence scores range from 1 to 4,
indicating the number of existing PF products that consistently classify each pixel as PF.




Despite achieving an acceptable accuracy, the maps of NF and PF contain inherent uncertainties.
Firstly, the forest samples and post-processing step are determined using a series of forest masks.
Uncertainties in these forest masks affect the precise mapping of NF and PF. Secondly, the training
samples also suffer from some mislabeled issues, as the frequency of disturbances between PF and NF
often leads to confusion (Xiao et al., 2024). For example, samples from NF areas with frequent fires and
PF areas with no disturbance between 1990 and 2020 are more susceptible to mislabeling. Additionally,
there is a continuous transformation process from PF to NF. Specifically, unmanaged PF gradually
transforms into a semi-natural state and eventually into NF, resulting in no clear boundary between
unmanaged PF and secondary forests. Although the sampling strategy, the low proportion of such areas,
and the use of relatively conservative (i.e., low-sensitivity) parameters in CCDC help mitigate the
impact of mislabeled samples, errors are still introduced into the proposed method. Some studies
demonstrated that the probabilistic RF classifiers are suitable for classification when the training sample
set contains few mislabeled training samples (Htitiou et al., 2021; Belgiu and Csillik, 2018; Reis et al.,
2018; Wang and Jia, 2009). Implementing a soft classification strategy is recommended to enhance the
accuracy when the target is in fragmented areas (Pan et al., 2012), which matches the situation of
China's small-scale landscapes. Thus, future work should focus on developing models that are capable
of tolerating mislabeled samples in training datasets. Moreover, the accuracy results across different
periods suggest a potential decline in classification performance when using training samples from 2020
to represent earlier years. This finding underscores the temporal limitations of static training datasets.
Future work may benefit from incorporating multi-year training samples into model development,
particularly if high-quality reference data from earlier periods becomes available, to improve
classification accuracy over time.
Additionally, although the C-PFM method provides a robust approach for detecting forest changes,
some omission and commission errors in identifying stand-replacement disturbances are inevitable due
to temporal fitting errors caused by data gaps and image noise. In this study, the CCDC algorithm was
employed as the temporal segmentation algorithm owing to its capability to utilize all available Landsat
observations without requiring compositing. This maximizes the exploitation of temporal information
and enhances the detection of forest changes. Moreover, two critical parameters in CCDC,
*chiSquareProbability* and *minObservations*, were configured following Xiao et al. (2023) to specifically
target stand-replacement disturbances. These settings help reduce commission errors and improve the
reliability of PF mapping. However, uncertainties persist, particularly in areas affected by inconsistent
data availability and image noise. Future research may address these limitations by integrating

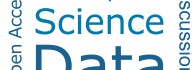

complementary remote sensing datasets from other platforms, such as MODIS and Sentinel-2, to
increase temporal coverage and spectral diversity. Additionally, the utilization of auxiliary datasets and
additional features has the potential to enhance the detection of forest changes. Furthermore, the
introduction of disturbance-sensitive indices, such as the Normalized Difference Fraction Index (NDFI),
has proven effective in improving classification accuracy in areas dominated by mixed pixels (Chen et
al., 2021).

## 6 Data availability

The produced 30 m annual maps for PF and NF in China are openly available at
https://doi.org/10.5281/zenodo.15559086 (Xiao, 2025).

## 7 Conclusion

This study introduces the C-PFM approach to monitor annual NF and PF dynamics at a 30 m spatial
resolution in China from 1990 to 2020. The annual maps produced by the C-PFM method demonstrated
reliability, supported by satisfactory accuracy when validated against visually interpreted reference data,
and acceptable Pearson's product-moment correlations with the 5th to 9th NFI data. Our findings reveal
that PF has predominantly driven the increase in forest cover in China, underscoring the significant
impact of afforestation initiatives, particularly in programs such as the upper and middle reaches of
Yangtze River Shelterbelt Program. This suggests that the targeted afforestation strategies in these areas
may be particularly effective. These insights are vital for shaping China's policies and initiatives that
aim at achieving carbon neutrality. Overall, the methods and data produced in this research provide a
solid foundation for further scientific investigation and policy development, enhancing our
understanding of forest expansion mechanisms and their implications for environmental conservation.

## Author contributions

YX designed the research, analyzed the data, wrote the original manuscript, and produced the dataset.
QW revised the whole manuscript and provided funding to support the research.

## Declaration of Competing Interest

The authors declare that they have no known competing financial interests or personal relationships



that could have appeared to influence the work reported in this paper.
**Acknowledgment**
This research was supported by the National Natural Science Foundation of China under Grants
42222108 and 42171345.

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
