# Peer review of "Monitoring planted forest expansion from 1990-2020 in China"

_Earth System Science Data, 2025_

## Community Comment (CC1)

**Comment on : Monitoring planted forest expansion from 1990-2020 in China**

1. Core advantages

Innovative and practical method: Proposed the C-PFM method (CCDC+RF), which automatically generates 1.34 million training samples, integrates multiple sources of high confidence regions and disturbance frequency discrimination samples, and reduces the limitations of traditional manual sampling; Generate annual PF/NF maps with a resolution of 30 meters from 1990 to 2020, breaking through the limitation of 5-year intervals in similar research and accurately capturing dynamic processes.

Complete verification system: In 2020, the map OA reached 90.8%, the F1 score of PF was 79.2%, and the Pearson correlation coefficient with NFI data was high, showing better performance compared to four existing products; Conduct spatial stratification analysis based on climate zones and forest types to avoid regional bias.

Outstanding application value: Quantify the net increase of 16.15 Mha in national PF and the net decrease of 8.09 Mha in NF, clarify the expansion focus of PF in the engineering area, and provide a basis for carbon neutrality; The data is fully publicly available through Zenodo, in line with the trend of open science.

2. Main shortcomings

Sample and classification uncertainty: There are insufficient samples in ecologically fragile areas (such as Xinjiang and Qinghai), with 27.2% of PF pixels only recognized by C-PFM; Without introducing dynamic features to distinguish semi natural forests, the boundaries are fuzzy and prone to misclassification.

Weak model generalization: The local RF model has a validation accuracy of only 62.3% -73.8% in new regions; The robustness of CCDC core parameters has not been analyzed.

Shallow analysis of driving and ecology: Without combining policy nodes to quantify the contribution of policies to PF expansion; Only qualitative mention of PF ecological impact, lacking quantitative analysis.

Insufficient early validation: From 1990 to 2000, the sample size was small (89) and could not reflect the accuracy of early classification.

3. Suggestion

Optimize samples and classification: supplement fragile area samples and use active learning to optimize semi natural forest labeling; Add dynamic features (such as disturbance frequency change rate) to distinguish semi natural forests.

Improve model generalization: Build a "global local joint training" model to supplement climate and terrain features; Conduct sensitivity experiments on CCDC parameters.

Deepening analysis: Consider using the DID model to quantify policy effects; Quantitatively evaluate the ecological benefits of PF based on GEDI and GRACE data.

Improve early validation: Integrate Landsat 5 imagery and provincial forestry data to supplement early samples.